

# A simulator for the CLARA-A2 cloud climate data record and its application to assess EC-Earth polar cloudiness

Salomon Eliasson[1], Karl-Göran Karlsson[1], and Ulrika Willén[1]

[1]Swedish Meteorological and Hydrological Institute, Folkborgsvägen 17, 601 76 Norrköping, Sweden

**Correspondence:** Salomon Eliassson (salomon.eliasson@smhi.se)

**Abstract.**

This paper describes a new satellite simulator for the Satellite Application Facility on Climate Monitoring (CM SAF) cLoud, Albedo and RAdiation dataset (CLARA), Advanced Very High Resolution Radiometer (AVHRR)-based, version 2 (CLARA-A2) Climate Data Record (CDR). This simulator takes into account the variable skill in cloud detection in the CLARA-A2 CDR by using a different approach to other similar satellite simulators to emulate the ability to detect clouds.

In particular, the paper describes three methods to filter out clouds from climate models undetectable by observations. The first method, compared to the simulators in Cloud Feedback Model Intercomparison Project (CFMIP) Observation Simulator Package (COSP), relies on one global visible cloud optical depth at 550nm ($\tau_c$) threshold to delineate cloudy and cloud-free conditions. Method two and three apply long/lat -gridded values separated by day and nighttime conditions. Method two uses gridded varying $\tau_c$ as opposed to method one that uses just a $\tau_c$ threshold, and method three uses a cloud Probability of Detection (POD) depending on the model $\tau_c$.

Method two and three replicate the relative ease or difficulty for cloud retrievals depending on the region and illumination by increasing the cloud sensitivity where the cloud retrievals are relatively straightforward, such as over mid-latitude oceans, and by decreasing the sensitivity where cloud retrievals are notoriously tricky, such as over the Arctic region during the polar night. Method three has the added advantage that it indirectly takes into account that cloud retrievals in some areas are more likely than others to miss some clouds. This situation is common in cold regions where even thick clouds may be inseparable from cold, snow-covered surfaces and also in areas with an abundance of broken and small scale cumulus clouds such as the atmospheric subsidence regions over the ocean.

The simulator, together with the International Satellite Cloud Climatology Project (ISCCP) simulator of COSP, is used to assess Arctic clouds in the EC-Earth climate model compared to the CLARA-A2 and ISCCP-H CDRs. Compared to CLARA-A2, EC-Earth is shown to underestimate cloudiness in the Arctic generally. However, compared to ISCCP and its simulator, the opposite conclusion is reached. Previous studies have found that the CLARA-A2 CDR performs well in the Arctic during the summer months, and this paper shows that the simulated cloud mask of CLARA-A2 using method three is more representative of the CDR than method one used in COSP, using a global $\tau_c$ threshold to simulate clouds. Therefore, the conclusion that EC-Earth underpredicts clouds in the Arctic is the more likely one.

The simulator substantially improves the simulation of the CLARA-A2 detected clouds, especially in the polar regions, by accounting for the variable cloud detection skill over the year. The approach to cloud simulation based on the POD of clouds





depending on their $\tau_c$, location, and illumination is the preferred one as it reduces cloudiness over a range of cloud optical depths. Climate model comparisons with satellite-derived information can be significantly improved by this approach, mainly by reducing the risk of misinterpreting problems with satellite retrievals as cloudiness features.

# 1 Introduction

Clouds constitute one of the most significant sources of uncertainties for projecting the future climate (IPCC, 2014). Therefore, countless studies have been made testing and improving the skill of climate models in this regard over the years (e.g., Waliser et al., 2009). As more and more information on cloud climatologies from satellite sensors are available in CDRs, climate models have been able to improve their representation of clouds continuously, and hence their description of the climate system itself.

To date, there are a few CDRs derived from imaging sensors that span more than 30 years; the ISCCP CDR (Young et al., 2018) which is based on cloud data from geostationary satellites, the Pathfinder Atmospheres- Extended (PATMOS-x) (Heidinger et al., 2014) and the Cloud_cci (Stengel et al., 2017) CDRs both based on measurements from the AVHRR instrument onboard National Oceanic and Atmospheric Administration (NOAA) and Meteorological Operational Satellite (MetOp) polar-orbiting satellites, and the CLARA-A2 CDR (Karlsson et al., 2017) which is also based on AVHRR data and contains 34 years of cloud data to date. The long length of these CDRs make them ideal for assessing the cloud climatologies of climate models.

However, to directly compare model clouds to cloud observations from satellites is akin to comparing "apples to oranges" as is explained in Waliser et al. (2009); Eliasson et al. (2011), and many others. Two of the primary considerations to make when comparing climate models to satellite observations is their very different horizontal and vertical scales, and the observations' finite sensitivity to clouds. Therefore, nowadays, in order to utilize the CDRs from satellite data, the CDRs usually need to be simulated from the model atmosphere with these attributes/limitations in mind.

In general, satellite simulators create cloud products or brightness temperatures that would have been made from satellite measurements if the model atmosphere was the real atmosphere. The simulators' objective is to emulate the inherent limitations, sensitivity, and geometry of the real retrievals. One of the main tasks for these simulators, among others, is to filter out model clouds that would not be detected by the instrument behind the cloud CDR. These simulated satellite products can then be compared directly to the observations.

Satellite simulators are primarily used to validate earth system models such as climate models. Although satellite simulators bridge the gap between models and observations by significantly reducing the comparison uncertainties, they do not eliminate them, and this should be taken into account when comparing satellite product simulations to the observations (Pincus et al., 2012). This paper introduces the CLARA-A2 satellite simulator v1.0, for use in model validations compared to the CLARA-A2 CDR.

The COSP (Bodas-Salcedo et al., 2011; Swales et al., 2018) was developed to gather and provide a suite of satellite simulators. These simulators provide column-integrated cloud retrievals, just as the datasets they represent, and therefore they need the cloud averages on the coarse grid of climate models to be translated into many smaller subcolumns for each model long/lat-grid box (Jakob and Klein, 1999; Pincus et al., 2006). The number of subcolumns per grid depends on the host models' reso-





lution, and typically number around $100\times$ the model resolution in degrees. Therefore, if a model has a resolution of $0.7°$, the simulator will generate 70 subcolumns per horizontal grid. As described in Jakob and Klein (1999), the subcolumns in a grid produce a horizontal cloud distribution, and each subcolumn has a cloud vertical structure determined according to the cloud overlap assumptions of the host model. The cloud retrieval simulations are further carried out on each of these subcolumns.

The ISCCP (Jakob and Klein, 1999), the MODerate resolution Imaging Spectroradiometer (MODIS) (Pincus et al., 2012), and the Multi-angle Imaging SpectroRadiometer (MISR) simulators are the visible/infrared (VIS/IR) satellite dataset simulators in COSP. The CLARA-A2 cloud products are also retrieved using an instrument that measures in this frequency range, and hence the CLARA-A2 simulator has many similarities with these. Other VIS/IR satellite simulators not included in COSP to date are the Spinning Enhanced Visible Infrared Imager (SEVIRI) (Bugliaro et al., 2011) and the Cloud_cci (Eliasson et al.,
2019) simulators.

    All satellite datasets based on VIS/IR data have regionally varying skill in detecting clouds, and all retrievals suffer when clouds are too tenuous to detect, or obscured. The removal of would-be undetectable clouds from the model is an essential feature of satellite simulators and to date is being carried out by comparing the $\tau_c$ of a subcolumn to some threshold value. To date, the simulators in COSP and the Cloud_cci simulator rely on a global static $\tau_c$ value to reclassify subcolumns, with an op-
tical depth less than this threshold, as cloud free. It is well established that all cloud masks based on the AVHRR channels have a variable skill, mainly depending on the underlying surface and the illumination conditions (e.g., Karlsson and Håkansson, 2018). Karlsson and Håkansson (2018) studied the performance of the CLARA-A2 cloud mask against Cloud-Aerosol Lidar with Orthogonal Polarisation (CALIOP) measurements in detail and produced global statistics for different $\tau_c$ thresholds, the probability of cloud detection, and the rate of falsely detected clouds (false alarm rate), on a global and regional basis. For
instance, they showed that the general likelihood of detecting clouds is much higher over warm ocean surfaces than over perpetually ice-covered regions and likewise that in some regions, e.g., deserts and other dry surfaces, retrievals there are relatively susceptible to producing false clouds.

    It is clear that the use of a fixed $\tau_c$ threshold, applied globally to modeled cloud fields in order to simulate satellite-based cloud detection limitations, is a substantial simplification of the actual observation conditions. Therefore a completely new
approach is introduced in this paper describing a simulator for the CLARA-A2 CDR applying spatially and temporally varying cloud detection thresholds. Employing this novel approach to simulating observed cloud cover, should put the confidence in cloud cover comparisons between the climate models and the CLARA-A2 CDR on a stronger footing. The CLARA-A2 simulator also incorporates a method of model temporal sampling in order to reduce errors potentially introduced by not taking the different and changing equatorial overpass times of the satellites used in the CLARA-A2 CDR, into account. This approach
is also used in the Cloud_cci simulator and is motivated and described in Eliasson et al. (2019).

    The article structure is as follows: Sect. 2.1, Sect. 2.2, and Sect. 2.3 describe the CLARA-A2 CDR, ISCCP-H series (ISCCP-H) CDR, and the EC-Earth climate model (Hazeleger et al., 2010) respectively. Section 3 describes the CLARA-A2 simulator and the simulated variables and Sect. 3.1 offers a description and demonstration of the simulated cloud masks. The CLARA-A2 simulator approach is demonstrated and tested over the Arctic region where trends in polar summer cloudiness is investigated
using simulations from the EC-Earth climate model in Sect. 4. A summary and conclusion is given in Sect. 5.





## 2 Data

### 2.1 The CLARA-A2 climate data record

The CLARA-A2 CDR is based on long term measurements from the AVHRR instrument operated onboard polar orbiting
NOAA satellites as well as onboard the MetOp polar orbiters operated by European Organisation for the Exploitation of
Meteorological Satellites (EUMETSAT) since 2006. AVHRR measures in five spectral channels (two visible and three infrared
channels) with an original horizontal field of view (FOV) resolution at the nadir of $1.1\,\mathrm{km}$. However, the data used in CLARA-
A2 is a reduced resolution ($5\,\mathrm{km}$) resampled version of these measurements, called global area coverage (GAC), where three
consecutive scanlines made up of 3x5 original FOVs make one GAC pixel. Specifically, the average radiance from four out of
five pixels from the first scan line and none from the next two scan lines are used to create the GAC measurement. Thus, only
about $27\,\%$ of the nominal GAC FOV is actually used (Karlsson and Håkansson, 2018, see Fig. 1). Only GAC data is available
globally (i.e., being archived) over the full period since the introduction of the AVHRR sensor in space.

The visible radiances were inter-calibrated and homogenized, using MODIS data as a reference before applying the multiple
parameter retrievals. The inter-calibration uses the method introduced by Heidinger et al. (2010), which is now updated using
MODIS Collection 6 as well extended by six years. The calibration of infrared AVHRR channels is based on the standard
NOAA calibration methodology utilizing an onboard blackbody reference (Rao et al., 1993). CLARA-A2 is an improved and
extended follow-up of the first version, CLARA, AVHRR-based, version 1 (CLARA-A1) of the record (Karlsson et al., 2013)
and is extended to cover 34 years (1982–2015).

CLARA-A2 features a range of cloud products: cloud mask (cloud amount), cloud top temperature/pressure/height, cloud
thermodynamic phase, and for liquid and ice clouds separately, cloud optical thickness, particle effective radius, and cloud water
path. Cloud products are available as monthly and daily averages in a $0.25°$ latitude-longitude grid and also as daily resampled
global products (Level 2b) on a $0.05°$ grid for individual satellites. The CDR also includes multi-parameter distributions (i.e.,
joint frequency histograms of cloud optical thickness, cloud top pressure, and cloud phase) for daytime conditions. Besides
cloud products, CLARA-A2 also includes surface radiation budget and surface albedo products. Examples of CLARA-A2
products can be found in Karlsson et al. (2017).

In this study, we focus exclusively on the AVHRR GAC cloud mask because of its central importance for the quality
of all other CLARA-A2 products. Validation results for other CLARA-A2 products can be found in Karlsson et al. (2017)
and CMSAF1 (2017). The method for generating the CLARA-A2 cloud mask originates from Dybbroe et al. (2005), but
significant improvements and adaptations since then were made to enable reliable processing of the historic AVHRR GAC
record (CMSAF2, 2017).

### 2.1.1 The skill of the CLARA-A2 CDR

As mentioned earlier, Karlsson and Håkansson (2018) performed an extensive validation of the CLARA-A2 cloud mask against
simultaneous nadir observations (SNOs) of CALIOP retrievals, and following is a recap of their main results. The goal was to
find out at which optical depth thin clouds were thick enough to have a $50\,\%$ probability of being detected. They investigated





the global performance of the CLARA-A2 cloud mask on a global equal-area grid with a 300 km resolution, covering different
surface types, and separately for daytime and nighttime conditions. This detection level can be considered the baseline for any
cloud mask; i.e., the smallest $\tau_c$ threshold where the cloud mask detects more clouds than it misses. They found that the global
mean minimum cloud optical thickness was $\tau_c = 0.225$. However, importantly, their results showed that the global mean is
far from being representative of all local conditions. For instance, a $\tau_c$ threshold value of 0.07 is a better approximation over
ice-free oceanic regions at mid-latitudes, whereas a $\tau_c$ threshold value as high as 4.5 is suitable for some ice-capped regions
such as over Greenland and Antarctica.

However, the capability of the cloud mask in CLARA-A2 is better described by the POD of clouds rather than a $\tau_c$ threshold.
Karlsson and Håkansson (2018) showed that even if all thin clouds with a $\tau_c$ less than 0.225 are removed from the comparison,
i.e., by reclassifying such CALIOP reference clouds as cloud-free, the POD varies considerably per region. Additionally, they
showed that for most regions in the world, the probability of detecting clouds with a $\tau_c$ near the average of 0.225 is higher than
50 % (Karlsson and Håkansson, 2018, see Fig. 9).

Through their validation studies, POD was calculated for $\tau_c$-intervals (or bins) based on these SNO validations on an equal-
area Fibonacci grid with about a 300 km radius. A Fibonacci grid is a type of grid where each grid box is nearly equal in size
and area (see Karlsson and Håkansson (2018) and references therein for more information). Figure 1 shows the different POD
for clouds that have an optical depth that falls in the optical depth interval centered around 0.225 ($0.2 < \tau_c < 0.25$) for daytime,
nighttime and all conditions. The figure shows that the POD of clouds in this optical depth range is dependent on whether
clouds are sunlit[1] or not, especially in the polar regions. The global average POD in this interval, but also all POD-intervals
(not shown), is somewhat skewed towards lower values due to the poor performance in the Polar regions during night time.
Another significant result in Fig. 1 is the exceptionally good POD results in the Arctic and Antarctic during the summer months.
CLARA-A2 has equal skill in detecting clouds in these regions during the sunlit months as it has over non-polar land regions.
This result further establishes the CLARA-A2 CDR as very suitable for cloud studies in the polar summer.

## 2.2 ISCCP-H

The ISCCP-H CDR (Young et al., 2018) is a recently released high resolution version of the ISCCP CDR (Rossow and
Schiffer, 1999) that starts in July 1983 and ends in June 2015 due to data availability at the time of this study. The ISCCP CDR
comprises of geostationary and polar-orbiting satellites, where data from the geostationary satellites have precedence at low
and mid-latitudes (absolute latitude $< 55°$). The main improvement of ISCCP-H CDR is that it is on a higher resolution spatial
grid compared to its predecessor and covers a longer period. Otherwise the ISCCP-H CDR is quite similar to previous ISCCP
versions. The CDR uses bi-spectral radiances, with one channel in the visible (0.6 µm) and one in the infrared (IR). This CDR
is described at more length in Karlsson and Devasthale (2018) and Tzallas et al. (2019).

---

[1]Sunlit refers to when the solar zenith angle is less than 80°.

**Figure 1.** Probability of detection of clouds having an optical depth between $0.2 \leq \tau_c < 0.25$. The $\tau_c$ in the center of this interval, 0.225, is the global average of the smallest $\tau_c$ threshold where the CLARA-A2 cloud mask detects more clouds than it misses according to CALIOP.(see text)

## 2.3 The EC-Earth model

160   The EC-Earth climate model (Hazeleger et al., 2010, 2012) is an earth system model with its atmospheric component based on the Integrated Forecast System (IFS) of the European Centre for Medium-Range Weather Forecasts (ECMWF). The version used for this study is 3.3, based on IFS cycle 36r4 is on a horizontal resolution of T255 with 91 vertical layers. The variant used in this study is the EC-Earth-Veg3 Atmospheric Model Inter-comparison Project (AMIP) simulation with prescribed monthly sea surface temperatures and sea ice conditions to enable comparisons with atmospheric observations. The temporal range used

165   to demonstrate the simulator covers 1982–2015 when compared only to the CLARA-A2 CDR and covers 198307–201506 when ISCCP-H is involved in the comparison. EC-Earth simulated ISCCP clouds at run time through the COSP.



**Table 1.** The cloud variables produced by the simulator. The middle column specifies the separate categories available for each variable, and the third column indicates under which illumination conditions the variables are available. The cloud liquid phase is not one of the output variables since each subcolumn in a model grid has a cloud phase so that the simulated cloud liquid phase would only be the average cloud water phase in the grid.

| Cloud variable | Categories | day/night |
|---|---|---|
| Cloud fraction | total, ice, liquid, low, mid and high | day and night |
| Cloud top | height, temperature, pressure | day and night |
| $\tau_c$ | liquid, ice | day only |
| cloud particle effective radius ($r_e$) | liquid, ice | day only |
| cloud water path (CWP) | liquid, ice | day only |
| cloud top pressure (CTP)-$\tau_c$ 2D histograms | liquid, ice | day only |

## 3 Description of the CLARA-A2 simulator

As briefly described in the introduction, the CLARA-A2 simulator relies on subcolumns within the climate model grid, as all COSP simulators do, to simulate the observational datasets' cloud variables. Each subcolumn is determined to be either cloud-free or cloudy, but in contrast to all of the COSP simulators and the Cloud_cci simulator, the CLARA-A2 simulator can be set to use one of the three cloud mask simulation methods described below. Simulations of cloud retrievals are performed on each cloudy subcolumn and represent the column-integrated retrievals of CLARA-A2.

For consistency with observations, if a cloud parameter requires sunlight for its' retrieval, it will only be simulated if the calculated solar zenith angle is less then $80°$. These include the cloud microphysical retrievals $\tau_c$, $r_e$, Water Path (WP), and the CTP-$\tau_c$ histograms.

In the same manner as the MODIS and Cloud_cci simulators, to retrieve $r_e$ and WP, the CLARA-A2 simulator uses pre-calculated lookup tables based on the same cloud microphysical models used in the CDR retrievals, in this case, CLARA-A2. The lookup tables were calculated based on the liquid water droplet properties retrieved using Mie theory for two-parameter gamma distributions[2] of spherical droplets (Rooij and van der Stap, 1984) and ice particle properties are retrieved by raytracing using a geometric optics approximation for imperfect, randomly oriented, hexagonal, 'mono-size' ice crystals (Hess et al., 1998).

Finally, after simulating the retrievals on each subgrid, the simulated cloud parameters are averaged to the climate model grid, so that they are ready to be directly compared to observations. Table 1 provides an overview of the simulated variables included in this simulator. The CLARA-A2 satellite simulator can currently only be run in an "offline"-mode, meaning that it relies on access to pre-processed model output files.

---

[2]The assumed effective variance is 0.15



## 3.1 Simulating CLARA-A2 cloud masks

As mentioned in Sect. 1, the main feature of the CLARA-A2 simulator is a more sophisticated simulation of the observational datasets' cloud mask. It is possible to choose between three methods of cloud mask simulation:

1) To use a global static $\tau_c$, and treat all subcolumns with a $\tau_c$ less than 0.225 as cloud free. This method is used for the equivalent simulators in COSP, except that they use a slightly higher $\tau_c$ threshold of 0.3.

2) To use gridded optical depth thresholds separately for day and night conditions.

3) Make use of the gridded POD for cloud retrievals separately for day and night conditions.

Following is a short description of these three approaches.

### 3.1.1 A globally static optical depth threshold

Method one is to simulate the cloud mask by using one global minimum cloud optical depth value. This is the classical approach used by the ISCCP, MODIS, MISR, and the Cloud_cci simulators. For the ISCCP, MODIS, and MISR simulators, this global limit is set to $\tau_c = 0.3$ (Pincus et al., 2012), and for the Cloud_cci simulator (Eliasson et al., 2019), to 0.2. As mentioned earlier, the global average $\tau_c$ threshold for the CLARA-A2 CDR is 0.225, and thus the threshold value used in method one of the CLARA-A2 simulator.

By the approach used in this method, $100\,\%$ of the cloudy subcolumns with an optical thickness less than the global average $\tau_c$ limit are treated as being cloud-free and $100\,\%$ of the subcolumns above this threshold are treated as cloudy. Since the threshold is a global average, this method does not consider the illumination conditions or the geographical location of the retrieval. The advantage of this approach is its robustness and simplicity. However, as mentioned in Sect. 2.1, this approach can lead to very misrepresentative cloud mask simulations in some geographical regions.

### 3.1.2 Gridded optical depth thresholds

The second method uses varying gridded optical depth thresholds. This method also relies on the robust and straightforward approach of reclassifying subcolumns with a small optical depth as cloud-free, while keeping those above this threshold cloudy. However, this method is designed to also take into account that the $\tau_c$-threshold, or cloud detection limit, varies geographically and depends on the solar illumination. This method relies on the gridded data that are used in Fig. 12 in Karlsson and Håkansson (2018) that shows the smallest $\tau_c$ threshold where the CLARA-A2 cloud mask detects more clouds than it misses (see Sect. 2.1).

Figure 2 shows the detection limits used in the simulator according to this method. As shown by the figure, the $\tau_c$ threshold varies quite strongly regionally and also depends on if the CLARA-A2 cloud mask can make use of solar channels or not. The global average $\tau_c$-threshold, included for reference in the figure, clearly shows that during sunlit conditions, the cloud mask is much more sensitive to thin clouds than a global average value of $\tau_c = 0.225$ suggests.

During sunlit conditions, the regions with the least cloud sensitivity are over the Arctic and the desert regions of the Sahara and Arabia, as well as a large patch in the central Pacific. During nighttime conditions, especially over the oceans, the cloud





## Day

## Night

**Figure 2.** The gridded cloud detection limit, i.e., the smallest $\tau_c$ threshold where the CLARA-A2 cloud mask detects more clouds than it misses according to CALIOP for sunlit (top) and nighttime (bottom) conditions. For reference, the global average $\tau_c$-threshold = 0.225 is shown as contour lines.





mask is generally less sensitive and is particularly degraded in the ice-covered regions. However, there is an improvement in cloud sensitivity in some regions during nighttime conditions. For instance, in the desert regions of Northern Africa and the Arabian Peninsula, and the worst performing areas in the central Pacific, the cloud mask is somewhat surprisingly better

than when these regions are sunlit. A more in-depth validation study on CLARA-A2 is provided in Karlsson et al. (2017) and Karlsson and Håkansson (2018).

Their results demonstrate that using two sets of gridded detection-limits gives a more realistic cloud mask, one for sunlit and one for nighttime conditions. Method two is more realistic than the global static minimum optical depth approach of method one (Sect. 3.1.1). However, the authors of this paper advocate the further improved simulated cloud mask based on the use of

PODs described in the next section that also emulates some of the expected variability in cloud detection over a range of cloud optical depths.

### 3.1.3 Probability of cloud detection

The third method is an approach to simulate the CLARA-A2 cloud mask using the POD, provided on a roughly $300\,\mathrm{km}$ grid, as a function of the cloud's optical thickness. These POD, discussed in Sect. 2.1, are treated as the likelihood that the cloud mask

would detect the model cloud given its optical thickness, geographical location, and whether or not it is sunlit. POD is reported separately for the set of $\tau_c$- intervals listed in Tab. 2. The main purpose of Tab. 2 is to list all of the POD intervals used to simulate the cloud mask, but it also provides a summary of average POD separated into Global, Ocean, Land outside the polar regions, and the Polar regions during sunlit conditions (nighttime in braces). As is completely intuitive, the POD increases for optically thicker clouds for all regions, and in general, the cloud mask is more sensitive to clouds over ice-free oceans.

Additionally, nowhere, and not even for the thickest clouds, does the POD reach 1. The reasons for this seeming paradox are discussed at length in Karlsson and Håkansson (2018), and here is a summary:

1. Thick clouds are likely undetectable if they have the same temperature as the underlying surface during nighttime conditions when solar reflectivity measurements are not available.

2. Collocation errors between CALIOP and AVHRR can cause a mismatch between the datasets. Some collocation error
is unavoidable due to the maximum time difference of 3 minutes, and that sometimes the geo-location data for AVHRR itself may not be sufficiently accurate.

3. Even if the data is ideally collocated, the FOVs of the measurements most likely differ somewhat due to how the GAC footprint is made (see Fig. 1 in Karlsson and Håkansson (2018) and Sect. 2.1 here).

In fairness, only the first point directly has to do with the skill of the CLARA-A2 cloud mask and thus should be simulated.
The next two bullets have to do with imperfections in the validation process, and therefore should not be simulated. Unfortunately, at this moment, the POD is reduced by all three points, and in the future, it could make sense to estimate and take into account the impact of all three of these considerations in the simulator.





**Table 2.** The probability of cloud detection for the CLARA-A2 cloud mask separated by intervals of CALIOP cloud optical thickness. This table shows the regional averages based on the POD values used in the simulator of large geographical regions. Note that the simulator makes use of gridded POD values on a $300\,\text{km}$ equal-area grid (see Fig. 3) and not the POD regional averages provided here for reference. The Polar region here refers to latitude > 75° N/S. The values apply to daytime (nighttime) conditions.

| $\tau_c$-centers | $\tau_c$-range | Global | Ocean | Land | Polar |
|---|---|---|---|---|---|
| 0.025 | $0.00<\tau_c\leq0.05$ | 0.31 (0.23) | 0.34 (0.32) | 0.30 (0.14) | 0.22 (0.08) |
| 0.075 | $0.05<\tau_c\leq0.10$ | 0.44 (0.29) | 0.49 (0.38) | 0.40 (0.22) | 0.33 (0.11) |
| 0.125 | $0.10<\tau_c\leq0.15$ | 0.49 (0.36) | 0.56 (0.47) | 0.43 (0.30) | 0.38 (0.13) |
| 0.175 | $0.15<\tau_c\leq0.20$ | 0.55 (0.43) | 0.62 (0.55) | 0.48 (0.38) | 0.43 (0.17) |
| 0.225 | $0.20<\tau_c\leq0.25$ | 0.59 (0.50) | 0.67 (0.63) | 0.51 (0.46) | 0.46 (0.20) |
| 0.275 | $0.25<\tau_c\leq0.30$ | 0.62 (0.56) | 0.70 (0.70) | 0.54 (0.52) | 0.49 (0.23) |
| 0.325 | $0.30<\tau_c\leq0.35$ | 0.64 (0.60) | 0.73 (0.75) | 0.57 (0.57) | 0.51 (0.25) |
| 0.375 | $0.35<\tau_c\leq0.40$ | 0.67 (0.64) | 0.75 (0.78) | 0.59 (0.61) | 0.53 (0.28) |
| 0.425 | $0.40<\tau_c\leq0.45$ | 0.69 (0.66) | 0.78 (0.81) | 0.62 (0.64) | 0.55 (0.30) |
| 0.475 | $0.45<\tau_c\leq0.50$ | 0.72 (0.68) | 0.80 (0.82) | 0.65 (0.66) | 0.58 (0.32) |
| 0.550 | $0.50<\tau_c\leq0.60$ | 0.74 (0.70) | 0.83 (0.84) | 0.68 (0.68) | 0.60 (0.34) |
| 0.650 | $0.60<\tau_c\leq0.70$ | 0.77 (0.72) | 0.85 (0.85) | 0.71 (0.70) | 0.62 (0.37) |
| 0.750 | $0.70<\tau_c\leq0.80$ | 0.79 (0.73) | 0.87 (0.85) | 0.74 (0.72) | 0.65 (0.39) |
| 0.850 | $0.80<\tau_c\leq0.90$ | 0.82 (0.74) | 0.89 (0.86) | 0.77 (0.74) | 0.67 (0.42) |
| 0.950 | $0.90<\tau_c\leq1.00$ | 0.84 (0.76) | 0.90 (0.86) | 0.80 (0.76) | 0.71 (0.47) |
| 1.500 | $1.00<\tau_c\leq2.00$ | 0.87 (0.78) | 0.92 (0.87) | 0.83 (0.79) | 0.76 (0.53) |
| 2.500 | $2.00<\tau_c\leq3.00$ | 0.90 (0.81) | 0.94 (0.89) | 0.87 (0.82) | 0.82 (0.59) |
| 3.500 | $3.00<\tau_c\leq4.00$ | 0.94 (0.84) | 0.97 (0.91) | 0.93 (0.86) | 0.88 (0.66) |
| 4.500 | $4.00<\tau_c\leq5.00$ | 0.97 (0.88) | 0.98 (0.93) | 0.96 (0.90) | 0.92 (0.70) |

On the other hand, results from Tab. 2 indicate that the impact of points two and three may not be that strong after all. Over global oceans during the daytime, where highest POD values are found, the detection rate for the most optically thick clouds

is 98 % indicating, on average, that the combined error from points two and three is probably less than 2 %. However, in some oceanic regions where relatively thick inhomogeneous clouds are prevalent, such as the stratocumulus-dominated regions off the west coast of South America and southern Africa, POD values are slightly below 0.9, hence the impact of points two and three may not be negligible in these regions.

To illustrate the global distribution of POD, Fig. 3 contrasts two $\tau_c$-intervals used by the simulator. Clouds that fall in the

interval centered at $\tau_c = 0.125$, which are translucent clouds at only half the global average $\tau_c$ -limit (see Sect. 3.1.1), generally have a low POD. The POD is particularly low in this interval over land and during nighttime conditions. However, take notice that especially over ocean areas and especially during sunlit hours there is at least a 50 % POD despite the clouds being so thin.

# Day

## τ0.125  τ0.55

# Night

## τ0.125  τ0.55

**Figure 3.** The probability of detection at two $\tau_c$-intervals centered at 0.125 and 0.55 for day and night conditions.

For clouds centered at $\tau_c = 0.55$, which is about twice the global average detection limit, the PODs are predictably quite high in general. However, again, this is not true globally. Even though the clouds are relatively thick, in areas such as Northern Africa, the Arabian peninsula, and the Polar regions, the POD is only around 50 %. Another striking feature is that for these semi-transparent clouds, the POD over nearly all regions, except the poles, are higher for cloud retrievals made during nighttime conditions. This result is demonstrated further in Fig. 4. Outside the polar regions, clouds in the $\tau_c$-intervals from 0.2 to 0.6 have a higher POD during nighttime conditions overall (especially in the Tropics), whereas for clouds thinner or thicker than this interval, the daytime cloud masks have better success.

That this slightly improved detectability at night for clouds in the $\tau_c$ range 0.5-1.0 is a robust feature is supported by intercomparisons made between CLARA-A2 and other AVHRR-based datasets (e.g., Karlsson et al., 2017; Karlsson and

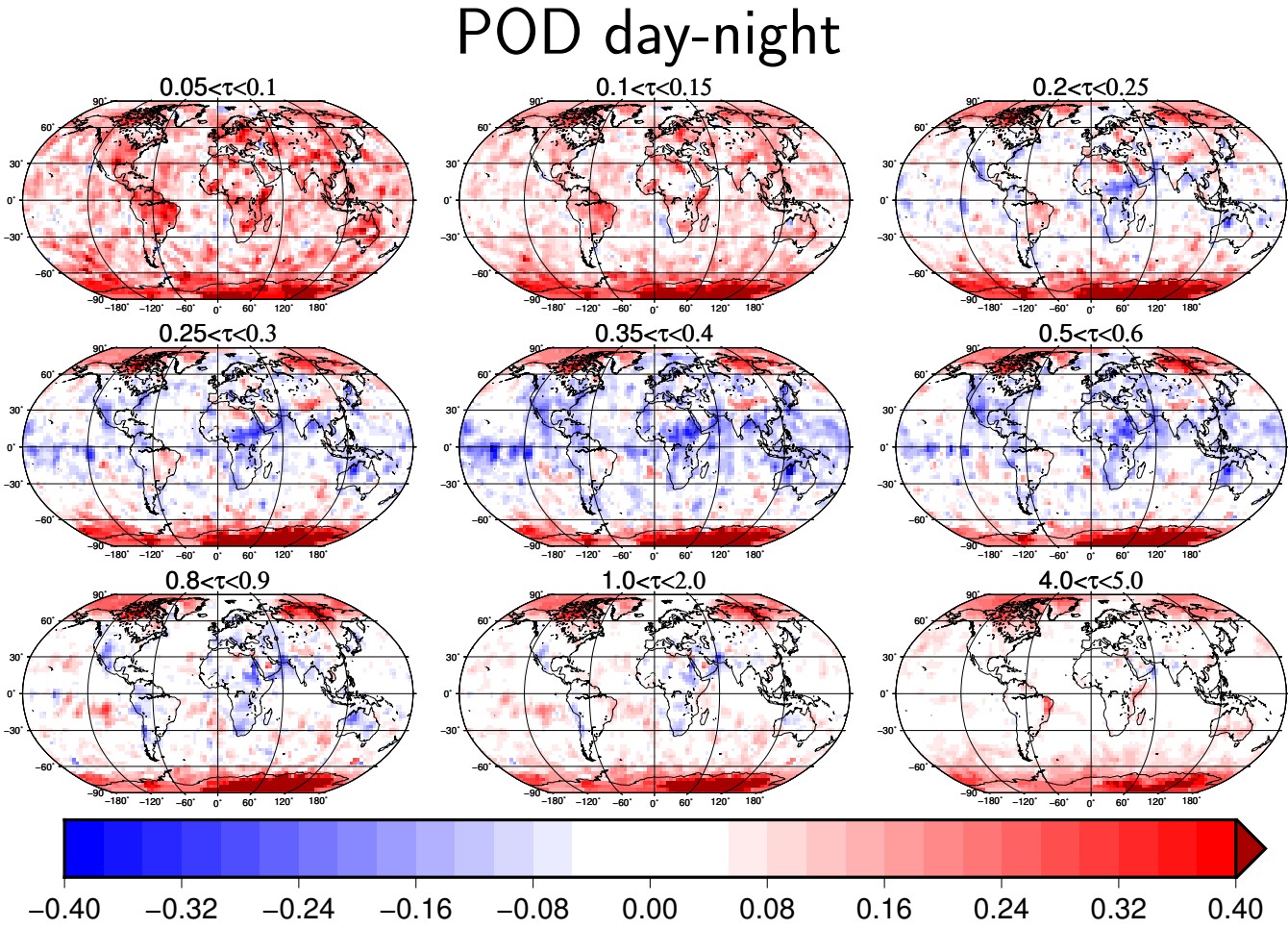

**Figure 4.** The difference in the POD of the cloud mask during sunlit and nighttime conditions for selected cloud optical depth intervals.

Devasthale, 2018). They found (although not explicitly reported in the papers) the same behavior for results from PATMOS-x and Cloud-cci compared to CALIOP observations. Whether to interpret this as an indeed improved nighttime detectability for AVHRR-based methods or something caused by the CALIOP observation reference (e.g., enhanced daytime problems due to

lower signal-to-noise ratios) is currently unclear. However, this feature is not critical to the CLARA-A2 simulator but merits a more in-depth investigation in the future.

The simulator uses computer-generated random numbers for comparison to the gridded POD value found in a lookup table, where one set of optical depth dependent- PODs is for sunlit, and one is for nighttime conditions. The simulator assigns a random number between 0–1 to each subcolumn at the initiation. After the simulated $\tau_c$ is computed, the column integrated $\tau_c$,

latitude, and longitude are used to find the POD value from the lookup table for comparison. A subcolumn is cloudy, only if its assigned random number is less than the POD. Therefore, if the probability of detection of a cloud with a specific optical depth is 0.05, even though it is very transparent, there is still a 5 % chance the subcolumn will be considered cloudy. Conversely, as





mentioned earlier, regardless of how optically thick a cloud is in a subcolumn, there is a non-zero chance this subcolumn will not be flagged as cloudy, and hence not included in any further cloud simulations.

### 280   3.2   The choice of simulated cloud mask

In this section we refer to figures 5 and 6 to illustrate how the choice of cloud mask simulation method affects the comparison of cloud cover of EC-Earth to CLARA-A2. The results are separated into seasons here since it is essential to understand the seasonal impact of choosing one method over another. The top figure in Fig. 5 and Fig. 6 show the cloud cover according to CLARA-A2 for 1982–2015 during Southern Hemisphere summers and the Northern Hemisphere summers respectively.

EC-Earth minus CLARA-A2 based on the first method (Sect. 3.1.1) is subplot (a), based on the second method (Sect. 3.1.2) is subplot (b), and based on the third method (Sect. 3.1.3) is subplot (c). Subplot (d) shows the difference between the simulated cloud mask based on method one, a global static $\tau_c$ threshold, and method three, based on POD thresholds (first method minus the third method).

Globally, the overall impression is that EC-Earth underestimates cloud fraction. In most regions of the world, within a few

percent, this is the conclusion one would reach regardless of which of the three methods was used to simulate the CLARA-A2 cloud mask. However, as described in Sect. 2.1, the CLARA-A2 CDR is systematically and substantially less skillful under certain conditions than on average.

As discussed in Sect. 2.1.1, CLARA-A2 is skillful at detecting clouds in the polar regions during sunlit conditions, but not so during the polar winter. This is why the apparent overestimation of clouds in these regions by EC-Earth (Fig. 5,6 a) is

likely strongly exaggerated. Without prior knowledge of the retrieval difficulties in cold dark locations, i.e., when only passive infrared channels are available, if method one is used to simulate clouds, one could erroneously conclude that EC-Earth places too many clouds in polar regions. This problem is especially salient during winter months, but it also has a considerable impact on cumulative averages over these regions. Therefore cloud mask simulations based on method one are notably unsuitable in the Polar regions and to lesser extent desert areas.

However, and what is the main point of this innovation, if one uses the second or third method to simulate clouds, the apparent bias in cloudiness in these regions is mostly removed in the problematic regions. The second and third methods do a much better job at reproducing the limitations of cloud datasets than the first method, and the size of the difference between method three and one is substantial and seasonally dependent in the problematic regions (Fig. 5,6 d).

Notice also from (Fig. 5,6 b) and (Fig. 5,6 c) that the second and third methods produce similar results, and hence both do

well in this regard. However, there are some subtle differences. One is that during the northern hemisphere summer months a model validation based on the second method leads to the conclusion that EC-Earth overestimates clouds in the Arctic, yet if the comparison were made based on the third method, one would conclude only a slight overestimation here.

The third method gives the most accurate description of the cloud detection limitations since it describes the likelihood of detecting/missing clouds over the full range of cloud optical thicknesses for day and night conditions. Also, method three can

emulate the non-zero probability that even thick clouds might be undetectable under certain conditions. This approach better describes the skill of the cloud retrievals of a satellite dataset than using gridded static values of $\tau_{\min}$ in method two, and



**Figure 5.** Total cloud cover during the DJF- seasons of 1982–2015. This figure shows a comparison of EC-Earth to CLARA-A2 using three different methods of cloud mask simulation. The reference figure at the top is the cloud fraction from CLARA-A2. Subfigure (a) shows the simulated observations using method one, based on a global static $\tau_c$-limit, minus CLARA-A2. Subfigure (b) shows the same comparison using method two, based on gridded $\tau_c$-limits, and (c) shows the same using method three, based on POD. Sub-figure (d) shows the difference between the simulated CF based on method one minus method three. See Sect. 3.2 for a wider description of the figure.





**Figure 6.** Total cloud cover during the JJA -seasons of 1982–2015. See the description in Fig. 5 for a description of the layout in this figure





especially instead of using a single global $\tau_{\min}$ value used by method one. Overall, therefore, the recommendation is to choose method three to simulate the cloud mask.

## 4 Application of the simulator to Arctic case studies

### 4.1 Average cloudiness during summer months

Karlsson et al. (2017) asserted, and the POD maps shown in Fig. 4 suggest, that the CLARA-A2 CDR is particularly skillful at detecting clouds in the Arctic during sunlit conditions. Therefore, to demonstrate the utility of the CLARA-A2 simulator, we assessed the cloud cover in these conditions over the full length of the datasets. We added the ISCCP-H CDR to the comparison since it is an equivalent CDR with a well-established satellite simulator used in many previous model studies (e.g., Webb et al.,
2001; Norris et al., 2016; Terai et al., 2016; Tan et al., 2017).

Figure 7 shows the average cloudiness in Arctic summer months according to CLARA-A2 (Fig. 7a) and ISCCP-H (Fig. 7b). EC-Earth's representation of overall cloudiness during Arctic summer is tested using the simulated CLARA-A2 and simulated ISCCP-H, shown in Fig. 7c and Fig. 7d respectively. The cloudiness from ISCCP-H should be compared to the simulated cloudiness using the ISCCP simulator, and the cloudiness, according to CLARA-A2, is compared to the CLARA-A2 simulator.
As mentioned in Sect. 3.1.1, the simulated cloud mask for ISCCP-H uses a global $\tau_c$ threshold ($\tau_c$=0.3) for the simulated cloud mask (method one, different threshold), and the CLARA-A2 simulator uses the POD-based approach for the simulated cloud mask (method three). All three datasets are limited to 198307–201506 to match the availability of the ISCCP-H period to date.

Fig. 7 demonstrates that using simulators that do not take the variable skill of the cloud mask into account, such as the ISCCP simulator, could easily lead to false conclusions about EC-Earth cloud cover in the Arctic summer. Compared to the ISCCP-H
observations, the simulated ISCCP-H observations indicate that EC-Earth has a strong positive cloud bias in the Arctic of about 40 % to 50 %. However, CLARA-A2, shown to have a high skill in the polar summer (see Fig. 5b Karlsson et al., 2017), indicates that EC-Earth under-predicts the cloudiness in this region by 20 % to 30 %. Similarly, Karlsson and Devasthale (2018) found the cloud cover of ISCCP-H too low in the polar summer and early autumn. The CLARA-A2 simulator shows that rather than EC-Earth massively overestimating cloudiness, especially over the central Arctic regions, EC-Earth has a similar amount
of clouds in the Arctic, and rather tends to under-represent clouds in the sunlit Arctic conditions. These large differences between the simulated ISCCP-H and CLARA-A2 are mainly due to the ISCCP simulator being too sensitive to thin clouds here. As shown in Fig. 2, during daytime conditions in the Arctic, a more appropriate daytime $\tau_c$-limit would be around 0.5 or more, which is higher than the global average of 0.3 assumed by the ISCCP simulator. Therefore in the Arctic summer, the ISCCP simulator retrieves clouds in between these cloud optical thicknesses that the CLARA-A2 simulator, and most likely
the observations, do not. As a consequence, anyone assessing cloudiness in the Arctic will reach the opposite conclusion using the CLARA-A2 CDR and simulator compared to the ISCCP-H counterpart.

Overall, based on CLARA-A2 as the reference, EC-Earth has a smaller average cloud fraction over most of the region between 50N–90N during the summer months. The difference is more substantial over ocean areas than over land, with the largest under-representation of cloudiness at these latitudes is over the North Atlantic and following the Gulf Stream north of



## Cloud Fraction JJA



**Figure 7.** The Total Cloud Fraction (TCF) in the Arctic summer. The top row contains the observations from two equivalent CDRs, CLARA-A2 (a) and ISCCP-H (b). The bottom row contains the simulated CDRs based on the model atmosphere of EC-Earth, simulated CLARA-A2 (c) and simulated ISCCP (d). The period is 198307–201506.





Norway. However, globally, the most considerable negative cloud biases between the model and observations are in the Tropics
and subtropics (see Fig. 6).

## 4.2    Trends in cloudiness

This section assesses trends in cloudiness during the Arctic sunlit months, according to CLARA-A2, and EC-Earth. The
CLARA-A2 CDR is suitable for cloud trend analysis since it is long enough to make statements about cloud trends and is
reported to have high cloud detection skills in the Arctic (Karlsson and Devasthale, 2018). Here is an assessment of the cloud
trends from the months that have enough sunlight, i.e., where the solar zenith angle is less than 80° in the Arctic above 70°.
These trends are decadal and based on the linear regression of cloudiness from all data in 1982–2015.

Fig. 8 shows the distribution of cloudiness trends, according to CLARA-A2. From this figure, some clear patterns emerge;
in the spring months, there is an increase in cloudiness by more than 5 % in large parts of the Arctic and upwards of 10 % north
of Novaja Zemlya, and in the summer to Autumn months the Arctic is dominated by a decrease in cloudiness. The increase in
cloudiness reaffirms observations previously reported in Kapsch et al. (2013) and Kapsch et al. (2019). Kapsch et al. (2013)
asserted that the increase in cloudiness is likely due to an increased intrusion of water vapor into these regions during the spring
months. The largest decrease in cloudiness seen in July and August is in the Beaufort, and especially the Lincoln Seas, north
of the Canadian archipelago and Greenland. However, it is outside the scope of this study, whose main purpose is to describe
the CLARA-A2 simulator, to further assess the possible reasons for the changing cloudiness seen in these observations.

Fig. 9 shows the average cloudiness trends for the same conditions, aside from excluding land areas, as in Fig. 8, for CLARA-
A2, the three methods of simulated CLARA cloud mask from the EC-Earth atmosphere, and the total cloudiness directly from
EC-Earth without any simulator. The reason for the simplified analysis in Fig. 9 is to avoid over-emphasizing differences in the
model cloudiness trends. EC-Earth is represented here by only one model run, and although it employs prescribed sea surface
temperatures and sea ice extent, the model atmosphere is free to meander. The implication is that a perfectly valid atmospheric
state based on one model is run hard to compare to observations fairly. In order to assess if the model cloud trends agree with
the observations, ideally, several ensemble model runs are required to find a general trend and to assess whether or not the
natural variability produced by the model is accurate (Koenigk et al., 2019).

However, Fig. 9 illustrates that regardless of which method is used to simulate cloudiness, or even using no simulator at all,
does not alter the average cloud trends in the Arctic. These results may indicate that the clouds in the model are not changing
the average range and distribution of optical thicknesses over time, even if the actual cloud amounts may change. However, no
definitive conclusions on model cloud trends in the Arctic can be drawn here for the reasons listed above, and a more thorough
examination of whether or not EC-Earth reproduces realistic cloud trends is also outside the scope of this study. Although the
choice of method does not appear at first glance to impact the model cloudiness trends, it still makes sense, in this case, to use
method three to simulate clouds, since it more closely reflects the skill of the CLARA-A2 dataset.



**Figure 8.** The average decadal cloudiness trend in the Arctic from the illuminated months of April to August according to CLARA-A2. Negative trends correspond to an average decrease in cloudiness over time. The trends are from all months in the period 1982–2015.

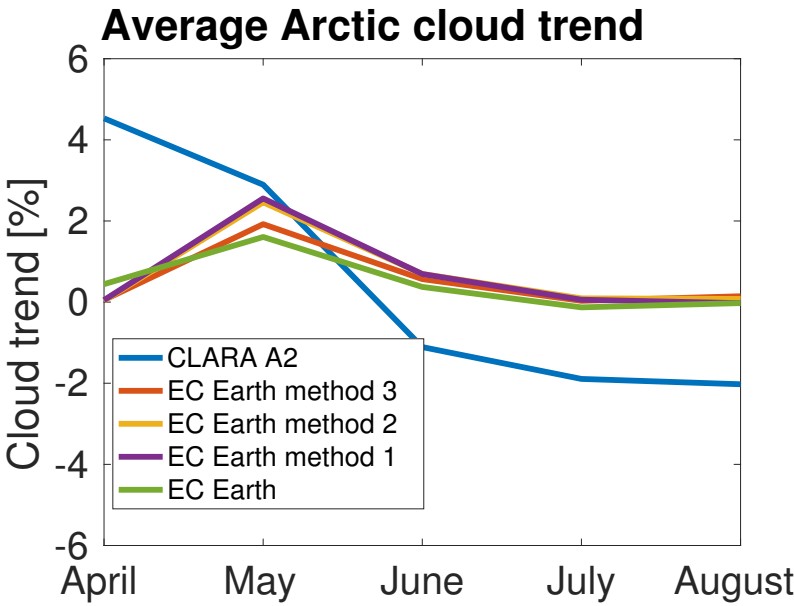

**Figure 9.** The average decadal cloudiness trend in the Arctic from the illuminated months of April to August only over the ocean (ice-free or ice-covered). The figure shows the reference dataset, CLARA-A2, the CLARA-A2 simulator, one line for each method, and the Total Cloud Cover (TCC) from the EC-Earth model without using any simulator. The trends are from all months in the period 1982–2015.

## 5    Conclusions

This article describes a satellite simulator designed to enable comparisons between climate models and the CLARA-A2 CDR. Typically, satellite simulators simulate the satellite retrieved cloud fraction using one global cloud optical depth threshold, called method one in this paper, to remove thin model clouds that are presumed undetectable by the instruments used to

generate the CDR. There are more factors to consider that influence the ability to retrieve thin clouds. These include

- – The optical thickness of the cloud

- – How illuminated the clouds are

- – The underlying surface properties and

- – The temperature difference between the cloud and the surface

In this paper, we show that using one optical depth value for all conditions to emulate cloud sensitivity (method one) is inappropriate since the cloud detection skill of satellite retrievals may vary considerably. This is the method used in some of the COSP simulators, which many studies have relied on, may have negatively impacted some model cloudiness analyses. There is a need for a more realistic simulated cloud mask that better reflects the actual cloud detection ability of the CDR.





We therefore propose two other methods that are both based on validations of the CLARA-A2 CDR using collocated cloud
retrievals from CALIOP by Karlsson and Håkansson (2018).

Method two uses two maps of cloud detection thresholds on a 300 km grid, one for day and one for night conditions.
These thresholds refer to the smallest cloud optical depth where there is a 50 % success rate in detecting clouds. The main
improvement by this method is that in areas where the cloud retrievals are relatively straightforward, such as over mid-latitude
oceans, the cloud sensitivity is generally increased, i.e., a lower cloud optical threshold. Conversely, in areas and conditions
where cloud retrievals are notoriously difficult, a much higher optical depth threshold is suitable.

Method three, the recommended approach to simulating the cloud mask, is based on the POD of clouds depending on their
$\tau_c$. Instead of using a $\tau_c$ threshold to determine whether or not a model cloud would have been detected, with this approach,
any model cloud could potentially be detected or missed. Maps of POD valid for separate optical depth ranges (see Tab. 2) are
used together with a random number generated at run time for every model subgrid column to determine cloudiness. These
are also provided on a 300 km grid and separated by day and night. The main improvement of this method is that it indirectly
takes into account that retrievals in some regions are more likely than others to miss thick clouds. This situation is common in
cold regions where thick clouds may be inseparable from cold snow-covered surfaces and also in regions with an abundance
of broken and small scale cumulus clouds such as the atmospheric subsidence regions over the ocean.

This paper illustrates that these new approaches to cloud mask simulation bring the model and observations much closer to
each other compared to using a fixed optical depth threshold globally to filter out clouds. They allow for a more realistic model
to satellite comparison, and thus reduces the likelihood that incorrect conclusions from model assessments are reached simply
due to cloud simulations not correctly representing the cloud retrievals of the CDR. Although methods two and three both
significantly improve cloud mask simulations, method three, using the POD approach, is better since it realistically mimics the
performance of the cloud mask of the CLARA-A2 CDR over the full range of cloud optical thicknesses.

The overall cloudiness in the Arctic during summer months from 1984–2014 is used to demonstrate the usefulness of
the simulator and the new approach to cloud mask simulation. The ISCCP-H CDR here complemented the comparison as a
second independent satellite dataset. Therefore, EC-Earth was assessed using both the ISCCP and CLARA-A2 simulators and
compared to the CDRs they should simulate. This comparison shows that EC-Earth seems to produce too few clouds in and
around the Arctic compared to CLARA-A2. However, despite the ISCCP-H CDR having more clouds than CLARA-A2 in
the Arctic summer months, compared to ISCCP-H and using the ISCCP simulator, the assessment on EC-Earth cloudiness
would lead to quite the opposite conclusion in some regions in the Arctic. The simulated ISCCP cloudiness is substantially
higher than the ISCCP observations. This overrepresentation of clouds is mostly due to the ISCCP simulator using a global
optical depth threshold that, in the Arctic is too generous. This example demonstrates the advantage of using the CLARA-A2
approach to cloud mask simulation compared to the traditional approach used by the ISCCP simulator and others. Although
only demonstrated in the Arctic summer in this paper, the POD approach, method 3, is also the most appropriate globally.

In terms of trends in overall cloudiness in the Arctic for all months with sunlit conditions from 1982–2015, the observations
from CLARA-A2 show a sharp increase in cloudiness over the years, especially in the ocean areas north of western Russia, in
the spring months of April and May. In the summer and early autumn months, there is a large area of decreasing cloudiness



in the seas just north of Canada and Greenland. Although only based on one model run, and therefore clear statements about
cloud trends in the model cannot be made, one can deduce that the average cloudiness trends from the model are very similar
using any simulator method, or no simulator at all.

In summary, the authors advocate an approach to cloud mask simulation based on the probability of detection of clouds
depending on their optical depth, location, and illumination. This study suggests that evaluations of climate model simulations
of cloudiness parameters would benefit substantially from using more advanced satellite simulators, which is a better way than
today, accounts for weaknesses and strengths of satellite retrievals.

*Author contributions.* Salomon Eliasson is the principal author. Karl Göran Karlsson provided data and expertise on the CLARA-A2 CDR.
Ulrika Willén provided expertise from the climate model perspective.

*Competing interests.* No competing interests

*Acknowledgements.* This work was partly funded by EUMETSAT in cooperation with the national meteorological institutes of Germany,
Sweden, Finland, the Netherlands, Belgium, Switzerland and the United Kingdom. And partly funded by the Swedish national space board
(Grant Agreement No. / Award No. 121/14).

*Data availability.* The CLARA-A2 CDR (Karlsson et al., 2017) can be downloaded from https://wui.cmsaf.eu (last access: 30 July 2019).
Data from the EC-Earth global climate model (Hazeleger et al., 2010) can be obtained from http://www.nextdataproject.it/?q=en/content/ec-
earth-cmip5-data-extraction (last access: 30 July 2019). The ISCCP-H (Young et al., 2018) products and other ISCCP products are available
from https://www.ncdc.noaa.gov/cdr/atmospheric/cloud-properties-isccp (last access: 30 July 2019)

*Code and data availability.* The simulator code itself is so far only available by contacting the authors.





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

**Appendix: Glossary**

$\tau_c$  visible cloud optical depth at 550nm. 1–3, 5–12, 14, 15, 17, 22

$r_e$  cloud particle effective radius. 7

**AMIP**  Atmospheric Model Inter-comparison Project. 5

**AVHRR**  Advanced Very High Resolution Radiometer. 1–4, 10, 12

**CALIOP**  Cloud-Aerosol Lidar with Orthogonal Polarisation. 3–6, 9–12, 21

**CDR**  Climate Data Record. 1–8, 14, 17–19, 21–23

**CFMIP**  Cloud Feedback Model Intercomparison Project. 1

**CLARA**  CM SAF cLoud, Albedo and RAdiation dataset. 1, 4, 19

**CLARA-A1**  CLARA, AVHRR-based, version 1. 4

**CLARA-A2**  CLARA, AVHRR-based, version 2. 1–15, 17–23

**CM SAF**  Satellite Application Facility on Climate Monitoring. 1

**COSP**  CFMIP Observation Simulator Package. 1–3, 6, 7, 21

