# Peer review of "A simulator for the CLARA-A2 cloud climate data record and its application to assess EC-Earth polar cloudiness"

_Geoscientific Model Development, 2019_

## Referee Comment (RC1) · Anonymous Referee #1 · 6 Sep 2019

Review of "A simulator for the CLARA-A2 cloud climate data record and its application to assess EC-Earth polar cloudiness", by Salomon Eliasson et al.

The manuscript presents a nice discussion of the impact of different techniques to compare climate model and satellite datasets with respect to cloudiness. They demonstrate that it is important to account for the illumination and surface characteristics dependent sensitivity of passive satellite methods to detect clouds which form the basis for several long-term climate records. The impact of simplifications commonly used in literature when comparing climate models and CDR is presented. This core objective of the paper – highly relevant for the community – is very well met and I suggest publication

after minor revision.

Material and results from earlier work are used quite extensively. Some examples of impact are somewhat repetitive, but overall this does not hamper readability and thus there is no need for shortening.

I basically only have one suggestion for improvement of the manuscript. The key content of the paper is the CLARA-A2 simulator. The whole description focuses on cloud detection. Other parameters as cloud top height, tau_c, r_e or WP are mentioned, but techniques are not described. If this is supposed to be the main reference for the CLARA-A2 simulator method, this should be extended. See my specific comment below.

Specific points:

Line 7, "compared to the simulators in CFMIP". It should probably read "comparable to the simulators in CFMIP". It took me a few more lines until I understood what the usual approach was. Please clarify.

Line 15, "Method three ...": Isn't this sentence just rewording the statement of the sentence before?

Line 23, "the simulated cloud mask of CLARA-A2": Please add "based on EC-Earth" for clarity.

Line 30: In the abstract I'm missing the information how the location-illumination dependent POD is found/ how the method is calibrated. Please add this information.

Line 104, "five pixels from the first scan line and none from the next two scan lines are used to create the GAC measurement.": Please explain why, with another sentence.

Line 105: Here you cite Figure 1 OF Karlsson and Hakansson 2018 and not Figure 1 IN THIS manuscript, right? Maybe "(Fig. 1 in Karlsson and Hakansson, 2018) " might be clearer.

Line 140: The same again. Better write "(Fig. 9 in ...".

Line 127 and 141: The use of the acronym "SNO" seems unnecessary. You just mention it twice and, at least for me, it's not a very common acronym and thus not easy to read.

Line 150: Can you please comment on the lowest tau detected by CALIOP and its impact on a comparison with the model clouds.

Line 157: Why "IR" instead of a wavelength? Are they different? Then please give a wavelength range. Line 165 and again in line 327, "198307–201506" Please change the date format to something more readable: E.g. "July, 1983 – June 2015"

Line 167, Section 3: On the first half page, I would expect a general layout of the simulator method. As I understood, the CLARA-A2 simulator is first presented in this manuscript and this will be the main reference for later use of it. You state that apart from cloud detection, cloud top height, tau_c, re, WP are produced by the simulator. The remaining section lays its focus on cloud detection only. Can you please extend the explanation a bit for the other parameters and how they are averaged? Starting from overlap assumption, subcolumns, and optical properties, the next step for a full simulator would be a radiative transfer forward step? Do you use this step to simulated satellite measured reflectivities? This could be the lookup table you mention, but it stays unclear. Where do you get r_e from? It can not be correctly derived by just averaging model columns (or subcolumns) vertically and horizontally in a simple way? Please extend description.

Fig. 2 and Fig. 3, 4 and Tab.2 are all results from earlier publications, aren't they (or at least based on them). This could be made more clear.

Line 272: It's only these last 6 lines of the section 3.1.3 which are not part of the summary based on Karlsson and Hakansson 2018, right? Think about pushing these lines into the next section as they clearly belong to the new retrieval simulator. They

are somewhat hidden here.

Line 323, "simulated ISCCP-H". Please give a reference again.

Line 327, "All three datasets ...": You just show two, don't you?

Line 327, hardly readable date format, as before

Line 331, "underpredicts cloudiness . . . by 20% to 30%": Can not be judged from the absolute images shown. Think about showing it in a similar way as in Fig 6.

Lines 335-341: This is basically all repetition, I think. Could be shortened in my opinion.

Typos/Language:

Lines 94/95: Should read "trends are inverstigated", "Summary and conclusion are given".

Line 366: "is run" → "run is"

―――――――――――――――――――

---

## Author Comment (AC1) · 18 Sep 2019

article color

Dear referee#1,

Thank you for taking the time to review our manuscript. We are especially happy that you agree that the core objective of the paper is clear and highly relevant for the community. Thank you also for the suggested minor revisions and following is a point by point response to each question/suggestion:

[Figure]

Line 7, "compared to the simulators in CFMIP". It should probably read "comparable to the simulators in CFMIP". It took me a few more lines until I understood what the usual approach was. Please clarify.

I have now reworded this sentence to "The first method, comparable to the simulators in COSP, relies on a single $\tau_c$- threshold applied globally to delineate cloudy and cloud-free conditions."

Line 15, "Method three ...": Isn't this sentence just rewording the statement of the sentence before?

I agree. I removed this sentence

Line 23, "the simulated cloud mask of CLARA-A2": Please add "based on EC-Earth" for clarity.

OK

Line 30: In the abstract I'm missing the information how the location-illumination de-pendent POD is found/ how the method is calibrated. Please add this information.

I have now added this sentence earlier on: "The gridded POD values are from the CLARA-A2 validation study by Karlsson and Håkansson (2018)"

Line 104, "five pixels from the first scan line and none from the next two scan lines are

used to create the GAC measurement.": Please explain why, with another sentence.

OK. I added this sentence to clarify the situation: "Saving the data on a GAC pixel resolution was a compromise to drastically reduce the data, a necessity due to limited bandwidth."

Line 105: Here you cite Figure 1 OF Karlsson and Hakansson 2018 and not Figure 1 IN THIS manuscript, right? Maybe "(Fig. 1 in Karlsson and Hakansson, 2018) " might be clearer.

Yes, I understand how this was confusing. Fixed

Line 140: The same again. Better write "(Fig. 9 in ...".

Fixed

Line 127 and 141: The use of the acronym "SNO" seems unnecessary. You just mention it twice and, at least for me, it's not a very common acronym and thus not easy to read.

I have remove them

Line 150: Can you please comment on the lowest tau detected by CALIOP and its impact on a comparison with the model clouds.

I added for reference that the optical depth sensitivity for CALIOP is about $\tau_c$=0.01 according to Winker et. al., 2009. The climate models have no lower limit besides numerical precision, i.e., much lower than CALIOP's lower limit. I also added a sentence about this in the paper to be clear

Line 157: Why "IR" instead of a wavelength? Are they different? Then please give a wavelength range. Line 165 and again in line 327, "198307–201506" Please change the date format to something more readable: E.g. "July, 1983 – June 2015"

I have included 11 micron and updated the date format

Line 167, Section 3: On the first half page, I would expect a general layout of the simulator method. As I understood, the CLARA-A2 simulator is first presented in this manuscript and this will be the main reference for later use of it. You state that apart from cloud detection, cloud top height, tau_c, re, WP are produced by the simulator. The remaining section lays its focus on cloud detection only. Can you please extend the explanation a bit for the other parameters and how they are averaged? Starting from overlap assumption, subcolumns, and optical properties, the next step for a full simulator would be a radiative transfer forward step? Do you use this step to simulated satellite measured reflectivities? This could be the lookup table you mention, but it stays unclear. Where do you get r_e from? It can not be correctly derived by just averaging model columns (or subcolumns) vertically and horizontally in a simple way? Please extend description.

OK, understood. I will expand this section to explain in much more detail how these other variables are simulated

Fig. 2 and Fig. 3, 4 and Tab.2 are all results from earlier publications, aren't they (or at least based on them). This could be made more clear.

Yes, the underlying results that are base for these figures and table where created for the Karlsson et.al., 2017 paper. I can add this information in the captions of the figures and see that it is clear in the text.

Line 272: It's only these last 6 lines of the section 3.1.3 which are not part of the summary based on Karlsson and Hakansson 2018, right? Think about pushing these lines into the next section as they clearly belong to the new retrieval simulator. They are somewhat hidden here.

I see your point, that this paragraph seems out of place. I think it may fit better at the beginning of this subsection before we go into detail about the $\tau_c$ intervals, illumination etc.

Line 323, "simulated ISCCP-H". Please give a reference again.

I changed the order of the sentences so that I can reference ISCCP-H again as well as the ISCCP simulator

Line 327, "All three datasets ...": You just show two, don't you?

By three datasets, I am referring to CLARA-A2, ISCCP-H, and EC Earth. For clarity I will write 2 satellite datasets and the climate model

Line 327, hardly readable date format, as before

Fixed

Line 331, "underpredicts cloudiness . . . by 20% to 30%": Can not be judged from the absolute images shown. Think about showing it in a similar way as in Fig 6

It's a good point. I can swap out the simulated datasets showing absolute cloudiness to showing absolute difference compared to the observation

Lines 335-341: This is basically all repetition, I think. Could be shortened in my opinion. Typos/Language:

It appears to me that especially lines 333-335 more or else repeat what is said in the information from lines 331-333. I removed the second duplicate and moved the sentence about ISCCP-H underestimating cloudiness to the earlier paragraph, and now I think it reads much better. Otherwise, to me, I think the latter information from lines 336-341 is necessary to explain why the ISCCP simulator produces more clouds in the Arctic summer that the CLARA simulator as seen in Fig. 7

Lines 94/95: Should read "trends are inverstigated", "Summary and conclusion are given".

Fixed

Line 366: "is run" → "run is"

Fixed

---

## Referee Comment (RC2) · Anonymous Referee #2 · 8 Oct 2019

This paper documents a new simulator for the CLARA-A2 record, and shows its potential to the study of polar cloudiness. It focusses on the main difference between this simulator and existing simulators: the method to derive the cloud mask. It introduces a new method that outperforms the simple optical depth threshold used in existing simulators, with the largest benefits felt in the polar regions. The paper shows that we now have an observational record and a simulator methodology that can be used to evaluate models in the polar regions with more confidence than before, and this is a very nice contribution to the model evaluation topic. The paper is well written, and I would recommend publication. I recommend some changes below. The changes requested are minor in the sense that they do not target any fundamental aspect of the

methodology, but moderate in the impact on the material presented.

GENERAL COMMENTS

Given that this paper will become the main documentation reference for this simulator, I think it would benefit from some discussion and results on the impact of the different methods (mainly #1 vs #3) on other variables listed in Table 1, not only cloud fraction.

The message regarding benefits of method #3 with respect to previous analyses needs to be more specific (e.g. in L385-390). The largest differences between methods occur in the polar regions, with much smaller differences in the rest of the globe. In some places, the paper gives the impression that previous studies where flawed, when in reality many of them did not use data polewards of 60 deg latitude to avoid large uncertainties.

Section 4.2. The observational pattern of trends is regionally inhomogeneous, and therefore Figure 9 is not very informative. Does EC-Earth show smaller trends due to compensation of regional patterns? It would be interesting to show the regional patterns from EC-Earth, perhaps replacing Figure 9 by a figure like Figure 8 but for EC-Earth.

SPECIFIC COMMENTS

- L36-41. This sentence is hard to read, please rewrite.

- L141-142. What's the difference between gridbox size and area?

- L148-150. This statement is slightly optimistic. Only subtropical deserts show PODs below 0.4 like most of the Arctic region. Most of the continental regions show larger PODs than the Arctic, and comparable or larger than the Antarctic region.

- Caption Table 1. Please can you clarify why the average cloud water phase is not a relevant quantity?

- L174. its' -> its.

- L187-183. There is no need to give details of the methods here, all that information is given in the subsections below.

- L200-205. It would be worth to point out that the COSP simulators only do the retrievals in sunlit conditions.

- Figure 2. The colour scale is very confusing, I would suggest a monotonic colour scale.

- Section 3.2. The POD maps used in method 3 depend on the distribution of clouds in the real world. These maps won't be optimal for models with cloud distributions that differ substantially from reality. It would be good to add a sentence mentioning this, and a brief discussion about the possibility of developing PODs that are not linked geographic positions.

- Figure 5 and 6. The labelling of the subplots is unusual. The top subplot should also have a label/letter so that it can be properly referenced.

- L352. The trends calculated in this section are not decadal trends. I believe that what you are trying to say is that they are trends over the entire record, expressed in units of %/decade.

- L354. Please use the correct units (%/decade). Same for figures 8 and 9. I would even suggest to change the units to 1/decade, as changes in % can lead to confusion in its interpretation (absolute percent change vs relative change).

- L366. is run -> run is

---

## Author Comment (AC2) · 1 Nov 2019

Dear referee#2,

Thank you for taking the time to review our paper for your overall supportive comments and useful suggestions about how to improve the article, and in particular, so that it may be more useful as a reference paper for the simulator. Following is a point by point response to each question/suggestion:

**General comments**

Given that this paper will become the main documentation reference for this simulator, we think it would benefit from some discussion and results on the impact of the different methods (mainly #1 vs #3) on other variables listed in Table 1, not only cloud fraction.

We agree that, since this will be the reference paper for the CLARA-A2 simulator, we should not only address the impact on cloud fraction of choosing the method of cloud cover simulation but also how the other cloud variables respond to the new method. We will add figures in a similar vein as the cloud fraction comparisons in figure 5 and fig 6 as well as an accompanying analysis.

The message regarding benefits of method #3 with respect to previous analyses needs to be more specific (e.g. in L385-390). The largest differences between methods occur in the polar regions, with much smaller differences in the rest of the globe. In some places, the paper gives the impression that previous studies where flawed, when in reality many of them did not use data polewards of 60 deg latitude to avoid large uncertainties.

Granted that, in this introductory paragraph in the conclusion section, we do not highlight the regionally variable impact of choosing a POD-approach (method 3) compared to using a static global optical depth threshold -approach (method 1). We will expand this paragraph to share the overall regional impacts, rather than just a global assessment as we do now.

We agree that the wording of this paragraph may also give the impression that there are many incorrect studies out there that have assessed simulated clouds in regions where it is inappropriate. This impression is not intentional, and we will make sure we are not implying this. We want to send the message that our approach can avoid sizable uncertainties.

We will mention that as long as model evaluations are carried out between +/- 60 degrees, and they usually are, the negative impact of using method one is not very large. However, we will also stress that by using a simulator that employs method 3, users need not limit their evaluation to +/- 60, especially not during the polar summer.

Section 4.2. The observational pattern of trends is regionally inhomogeneous, and therefore Figure 9 is not very informative. Does EC-Earth show smaller trends due to compensation of regional patterns? It would be interesting to show the regional patterns from EC-Earth, perhaps replacing Figure 9 by a figure like Figure 8 but for EC-Earth.

In the first version of the paper, we included the regional cloud trend patterns from EC Earth, but the results were tough to interpret. One of the problems is that we only have access to one realization of the model, and therefore no access to the model spread, which would be essential to assess cloud trends correctly here. Another, potentially worse problem is that the AMIP run of EC Earth used in this study, uses prescribed sea surface and ice cover surface conditions, and the sea-ice cover from ERA-Interim is fixed to 100% north of 80N for specific periods.
However, to make the comparisons of trends clearer, we will include the spatial distribution of cloud trends of EC-Earth trends instead of Figure 9. EC-Earth's trends also vary regionally; it is close to observations for some regions and not for others. We will comment on that in the text.

**Specific Comments**

L36-41. This sentence is hard to read, please rewrite.

We have split this super long sentence into several instead.

L141-142. What's the difference between gridbox size and area?

The Fibonacci grid is points spread approximately evenly over the globe, with the pixels matched to the closest point. The form is not quite round, nor is it a lat/long grid. To avoid confusion, we will remove the word 'size' and call it a "nearly equal-area grid".

L148-150. This statement is slightly optimistic. Only subtropical deserts show PODs below 0.4 like most of the Arctic region. Most of the continental regions show larger PODs than the Arctic, and comparable or larger than the Antarctic region.

We agree that the statement was too broad here. PODs in the polar regions are improving considerably during the polar summer (Day), but they are still not reaching values representative, e.g., most continental land surfaces. However, a strong point for the situation in the Polar summer is that if plotting a somewhat higher COT interval than shown here (e.g., 0.5-0.6), the differences decrease significantly between polar regions and most continental surfaces. This decrease is because of the higher skill in detecting liquid water clouds in the polar summer. The reason why this is not reflected in the current figure is that the very thin clouds in the COT interval 0.20-0.25 mostly consist of thin ice clouds, which are still difficult to detect over ice and snow surfaces in the polar summer.

Caption Table 1. Please can you clarify why the average cloud water phase is not a relevant quantity?

It may not be that the average cloud phase is irrelevant, but we have decided not to include this quantity. We suggest removing this confusing sentence.

- L174. its' → its.
OK

- L187-183. There is no need to give details of the methods here, all that information is given in the subsections below.

OK, we will remove this redundant text

- L200-205. It would be worth to point out that the COSP simulators only do the retrievals in sunlit conditions.

Thanks, we will point this out, and therefore also point out the advantage of this new simulator approach where the CLARA simulator can simulate cloud fraction and cloud top products all times of the year. We point out that the CLARA simulator does not produce COT, water path, or 2D CTP-COT histogram products during night time conditions

- Figure 2. The colour scale is very confusing, I would suggest a monotonic colour scale.

To us, the color scale is OK. However, we will change the top color from light pink to dark brown as a compromise, and hopefully, it will be less confusing

[Figure]

- Section 3.2. The POD maps used in method 3 depend on the distribution of clouds in the real world. These maps won't be optimal for models with cloud distributions that differ substantially from reality. It would be good to add a sentence mentioning this, and a brief discussion about the possibility of developing PODs that are not linked geographic positions.

We have to admit that we probably do not understand this question clearly. The CLARA-A2 simulator is a tool that should be used to facilitate model-to-satellite inter-comparisons and in this particular case, inter-comparisons with the results from the CLARA-A2 climate data record. So we are discussing clouds in the real world and not the cloud situation in a particular future or another scenario. If modeled clouds (channeled through the simulator) deviate from CLARA-A2 observations, it should be an indication of a model problem. This is the main goal for the simulator development. However, the reviewer is possibly asking how to interpret cases where models systematically place clouds incorrectly in space and then being subject to (potentially) other PODs than what they should have been in the CLARA-A2 simulator. The consequences here should not be large except for the extreme cases when a model place clouds over ice- and snow-covered areas in the polar night (with very low PODs) instead of over adjacent ice-free ocean areas (with very high PODs). Knowing about the unique problems over snow- and ice-covered regions (especially for the polar night) it will be hard to cover this situation adequately knowing about the specific cloud detection issues occurring over snow and ice during night conditions for AVHRR observations.

So, yes, under these particular circumstances, this might be a problem, and perhaps other observational datasets (e.g., from active sensors) would be more suitable to use here. However, for more normal situations, we do not believe this to be a big problem. Geographical mismatches between modeled and observed clouds should be possible to detect as long as the POD variability in the area of interest is not extreme.

We will add a brief discussion on this. We will use the example of the difficulties in

detecting clouds in the marginal ice zones and mention that as the ice margin moves in a warming climate, it will impact the POD geographical distributions. We plan on leaving the purely lat/long approach in future releases and base the PODs, preferably on something like climate zones or surface conditions.

- Figure 5 and 6. The labelling of the subplots is unusual. The top subplot should also have a label/letter so that it can be properly referenced.

Yes, the top subplot should be named (a). Also, subplot (d), soon to be (f), should be labeled "EC Earth (#1) - EC Earth (#3)" for clarity. We will change it accordingly.

L352. The trends calculated in this section are not decadal trends. I believe that what you are trying to say is that they are trends over the entire record, expressed in units of %/decade

Well, yes, this is what we are saying. We are using the wrong notation here and will fix it

- L354. Please use the correct units (%/decade). Same for figures 8 and 9. I would even suggest to change the units to 1/decade, as changes in % can lead to confusion in its interpretation (absolute percent change vs relative change).

We also accept to change the unit to 1/decade

- L366. is run → run is
OK

---

## Author Response (AR1)

Dear referee#1,

Thank you for taking the time to review our manuscript. We are especially happy that you agree that the core objective of the paper is clear and highly relevant for the community. Thank you also for the suggested minor revisions and following is a point by point response to each question/suggestion:

Dear referee#2,

Thank you for taking the time to review our paper for your overall supportive comments and useful suggestions about how to improve the article, and in particular, so that it may be more useful as a reference paper for the simulator. Following is a point by point response to each question/suggestion:

**General comments**

Reviewer 2: Given that this paper will become the main documentation reference for this simulator, we think it would benefit from some discussion and results on the impact of the different methods (mainly #1 vs #3) on other variables listed in Table 1, not only cloud fraction.

Answer: We agree that, since this will be the reference paper for the CLARA-A2 simulator, we should also describe all the simulated variables and not just the cloud fraction. We have expanded the section describing the simulator to also describe every variable in more detail, and decided that this is sufficient for the sake of a reference paper. In this paper, we want to focus on cloud fraction since it is absolutely central to this simulator.

Reviewer 2: The message regarding benefits of method #3 with respect to previous analyses needs to be more specific (e.g. in L385-390). The largest differences between methods occur in the polar regions, with much smaller differences in the rest of the globe. In some places, the paper gives the impression that previous studies where flawed, when in reality many of them did not use data polewards of 60 deg latitude to avoid large uncertainties.

Answer: Granted that, in this introductory paragraph in the conclusion section, we did not highlight the regionally variable impact of choosing a POD-approach (method 3) compared to using a static global optical depth threshold -approach (method 1). We have expanded this paragraph to share the overall regional impacts, rather than just a global assessment as we do now.

We agree that the wording of this paragraph also gave the impression that there are many incorrect studies out there that have assessed simulated clouds in regions where it is inappropriate. This impression is not intentional, and we have rewritten this paragraph to make sure that we are not implying this. We want to send the message that our approach can avoid sizable uncertainties.

We now mention that as long as model evaluations are carried out between +/- 60 degrees, and they usually are, the negative impact of using method one is

not very large. However, we also stress that by using a simulator that employs method 3, users need not limit their evaluation to +/- 60, especially not during the polar summer.

See the first paragraph of the conclusion in the marked-up version of the paper to see our reformulation

Reviewer 2: Section 4.2. The observational pattern of trends is regionally inhomogeneous, and therefore Figure 9 is not very informative. Does EC-Earth show smaller trends due to compensation of regional patterns? It would be interesting to show the regional patterns from EC-Earth, perhaps replacing Figure 9 by a figure like Figure 8 but for EC-Earth.

Answer: The EC-Earth pattern of trends is also regionally inhomogeneous. There is some cancellation between the regions but the main reasons for the smaller EC-Earth trend in Figure 9 is due to EC-Earth trends being smaller than observed, especially for the interior Arctic. We have now included the cloud trends for the climate model as we have for CLARA-A2. The results are, however, difficult to interpret. We describe in the text that we only have access to one realization of the model, and therefore no access to the model spread, which would be essential to assess cloud trends correctly here. We decided to keep Figure 9 in order to still raise the point that the choice of simulator does not seem to impact the cloudiness trend in the model.

Most of Sect 4.2 has changed to reflect this

**Specific comments**

Reviewer 1: Line 7, "compared to the simulators in CFMIP". It should probably read "comparable to the simulators in CFMIP". It took me a few more lines until I understood what the usual approach was. Please clarify.

Answer: We have now reworded this sentence to:

The first method, comparable to the simulators in COSP, relies on a single $\tau_c$-threshold applied globally to delineate cloudy and cloud-free conditions.

Reviewer 1: Line 15, "Method three ...": Isn't this sentence just rewording the statement of the sentence before?

Answer: We agree. We changed: "such as over the Arctic region during the polar night. Method three has the added advantage that it indirectly takes into account that cloud retrievals in some areas are more likely than others to miss some clouds. This situation is common in cold regions where even thick clouds may be inseparable from cold, snow-covered surfaces and also in areas" to

such as in cold regions at night, where thick clouds may be inseparable from cold, snow-covered surfaces, as well as in areas

Reviewer 1: Line 23, "the simulated cloud mask of CLARA-A2": Please add "based on EC-Earth" for clarity.

Answer: OK

Reviewer 1: Line 30: In the abstract I'm missing the information how the location-illumination dependent POD is found/ how the method is calibrated. Please add this information.

Answer: We have now added this sentence earlier on:

The gridded POD values are from the CLARA-A2 validation study by Karlsson and Håkansson (2018)

Reviewer 2: L36-41. This sentence is hard to read, please rewrite.

Answer: We have split this super long sentence into several instead:

Currently, there are only a few CDRs derived from imaging sensors that span more than 30 years. The ISCCP CDR (Young et al., 2018) was the first such dataset and mainly based geostationary satellite data, complemented with data from polar orbiting satellites at high latitudes. The three other CDRs are based on data from the polar-orbiting meteorological satellites from the National Oceanic and Atmospheric Administration (NOAA) and Meteorological Operational Satellite (METOP) series. They are the Pathfinder Atmospheres-Extended (PATMOS-x) (Heidinger et al., 2014), the Cloud_cci (Stengel et al., 2017), and the CLARA-A2 CDR.

Reviewer 1: Lines 94/95: Should read "trends are inverstigated", "Summary and conclusion are given".

Answer: Fixed

Reviewer 1: Line 104, "five pixels from the first scan line and none from the next two scan lines are used to create the GAC measurement.": Please explain why, with another sentence.

Answer: OK. We have added this sentence to clarify the situation:

Saving the data on a GAC pixel resolution was a compromise to drastically reduce the data, a necessity due to limited bandwidth.

Reviewer 1: Line 105: Here you cite Figure 1 OF Karlsson and Hakansson 2018 and not Figure 1 IN THIS manuscript, right? Maybe "(Fig. 1 in Karlsson and Hakansson, 2018) " might be clearer.

Answer: Yes, I understand how this was confusing. Fixed

Reviewer 1: Line 127 and 141: The use of the acronym "SNO" seems unnecessary. You just mention it twice and, at least for me, it's not a very common acronym and thus not easy to read.

Answer: I have remove them

Reviewer 1: Line 140: The same again. Better write "(Fig. 9 in ...".

Answer: Fixed

Reviewer 2: L141-142. What's the difference between gridbox size and area?

Answer: The Fibonacci grid is points spread approximately evenly over the

globe, with the pixels matched to the closest point. The form is not quite round, nor is it a lat/long grid. To avoid confusion, we removed the word 'size' and call it:

a nearly equal-area grid.

Reviewer 2: L148-150. This statement is slightly optimistic. Only subtropical deserts show PODs below 0.4 like most of the Arctic region. Most of the continental regions show larger PODs than the Arctic, and comparable or larger than the Antarctic region.

Answer: We agree that the statement was too broad here. PODs in the polar regions are improving considerably during the polar summer (day), but they are still not reaching values representative of most continental land surfaces. However, a strong point for the situation in the polar summer is that if plotting a somewhat higher COT interval than shown here (e.g., 0.5-0.6), the differences decrease significantly between polar regions and most continental surfaces. This decrease is because of the higher skill in detecting liquid water clouds in the polar summer. The reason why this is not reflected in the current figure is that the very thin clouds in the COT interval 0.20-0.25 mostly consist of thin ice clouds, which are still difficult to detect over ice and snow surfaces in the polar summer. We changed and expanded our statement to:

Another significant result in Fig. 1 is the high POD in the Arctic and Antarctic during the summer months. CLARA-A2 has nearly comparable skill in detecting clouds in these regions during the sunlit months as it has over non-polar land regions. Additionally, in the polar summer, for a somewhat higher COT interval than shown here (e.g., 0.5-0.6), the POD in polar regions increases more than most continental surfaces. This is due to a high skill in detecting liquid water clouds in the polar summer. The POD shown in Fig. 1 is somewhat lower here since clouds in the $\tau_c$ interval 0.20-0.25 mostly consist of thin ice clouds which are still difficult to detect over ice and snow surfaces. Overall though, this...

Reviewer 1: Line 150: Can you please comment on the lowest tau detected by CALIOP and its impact on a comparison with the model clouds.

Answer: OK. I added this sentence:

By comparison, the reference dataset, CALIOP can detect clouds with $\tau_c > 0.01$ (Winker et. al., 2009) and is generally stable across any surface.

Reviewer 1: Line 157: Why "IR" instead of a wavelength? Are they different? Then please give a wavelength range. Line 165 and again in line 327, "198307–201506" Please change the date format to something more readable: E.g. "July, 1983 – June 2015"

Answer: I have included 11 micron and updated the date format

infrared (11 µm)

July 1983 to June 2015

Reviewer 1: Line 167, Section 3: On the first half page, I would expect a general layout of the simulator method. As I understood, the CLARA-A2 simulator is first presented in this manuscript and this will be the main reference for later use of it. You state that apart from cloud detection, cloud top height, tau_c, re, WP are produced by the simulator. The remaining section lays its focus on cloud detection only. Can you please extend the explanation a bit for the other parameters and how they are averaged? Starting from overlap assumption, subcolumns, and optical properties, the next step for a full simulator would be a radiative transfer forward step? Do you use this step to simulated satellite measured reflectivities? This could be the lookup table you mention, but it stays unclear. Where do you get r_e from? It can not be correctly derived by just averaging model columns (or subcolumns) vertically and horizontally in a simple way? Please extend description.
Answer: Yes, this is clearly missing. This subsection is now rewritten to describe all the simulated variables (Section 3). Note, we choose to only shortly explain the simulation for the effective radius as this is described in detail in Pincus et. al., (2012). We reference also as such.

Reviewer 2: Caption Table 1. Please can you clarify why the average cloud water phase is not a relevant quantity?
Answer: It may not be that the average cloud phase is irrelevant, but we have decided not to include this quantity. We have removed this confusing sentence. In future versions of the CLARA simulator, we may decide to include it

Reviewer 2: - L174. its' → its.
Answer: OK

Reviewer 2: - L187-183. There is no need to give details of the methods here, all that information is given in the subsections below.
Answer: OK, we removed the numbered list

Reviewer 2: - L200-205. It would be worth to point out that the COSP simulators only do the retrievals in sunlit conditions.
Answer: Thanks, we now point this out as well as pointing out the added advantage of this new simulator approach. That is, we added that the CLARA simulator can simulate cloud fraction and cloud top products all times of the year. We also point out that the CLARA simulator does not produce COT, water path, or 2D CTP-COT histogram products during night time conditions

Reviewer 1: Fig. 2 and Fig. 3, 4 and Tab.2 are all results from earlier publications, aren't they (or at least based on them). This could be made more clear.
Answer: Yes, the underlying results that are base for these figures and table where created for the Karlsson et.al., (2018) paper. I can add this information in the captions of the figures and see that it is clear in the text.

Answer: To us, the color scale is OK. However, we changed the top color from violet to dark brown as a compromise, and hopefully, it will be less confusing

Answer: I see your point, that this paragraph seems out of place. I think it may fit better at the beginning of this subsection before we go into detail about the $\tau_c$ intervals, illumination etc.

Answer: We have to admit that we probably do not understand this question clearly. The CLARA-A2 simulator is a tool that should be used to facilitate model-to-satellite inter-comparisons and in this particular case, inter-comparisons with the results from the CLARA-A2 climate data record. So we are discussing clouds in the real world and not the cloud situation in a particular future or another scenario. If modeled clouds (channeled through the simulator) deviate from CLARA-A2 observations, it should be an indication of a model problem. This is the main goal for the simulator development.

However, the reviewer is possibly asking how to interpret cases where models systematically place clouds incorrectly in space and then being subject to (potentially) other PODs than what they should have been in the CLARA-A2 simulator. The consequences here should not be large except for the extreme cases when a model place clouds over ice- and snow-covered areas in the polar night (with very low PODs) instead of over adjacent ice-free ocean areas (with very high PODs). Knowing about the unique problems over snow- and ice-covered regions (especially for the polar night) it will be hard to cover this situation adequately knowing about the specific cloud detection issues occurring over snow and ice during night conditions for AVHRR observations.

So, yes, under these particular circumstances, this might be a problem, and perhaps other observational datasets (e.g., from active sensors) would be more suitable to use here. However, for more normal situations, we do not believe this to be a big problem. Geographical mismatches between modeled and observed clouds should be possible to detect as long as the POD variability in the area of interest is not extreme.

We added a brief discussion on this at the end of section 3.2. We plan on leaving the purely lat/long approach in future releases and preferably base the PODs on something like climate zones or surface conditions.

Answer: Yes, the top subplot should be named (a). Also, subplot (d), soon to be (e), should be labeled "EC Earth (#1) - EC Earth (#3)" for clarity. The figures have now been updated.

Answer: I changed the order of the sentences so that I can reference ISCCP-H again as well as the ISCCP simulator

Answer: By three datasets, I am referring to CLARA-A2, ISCCP-H, and EC Earth. For clarity we wrote:
The two satellite datasets and the climate model are limited to ...

Answer: Fixed

Answer: This is a good point. We swapped out the simulated datasets showing absolute cloudiness to showing absolute difference compared to the observations. After reexamining the data, we updated the stated biases since they seemed exaggerated. However, the message stayed the same

Reviewer 1: Lines 335-341: This is basically all repetition, I think. Could be shortened in my opinion. Typos/Language:

Answer: It appears to me that especially lines 333-335 more or else repeat what is said in the information from lines 331-333. I removed the second duplicate and moved the sentence about ISCCP-H underestimating cloudiness to the earlier paragraph, and now I think it reads much better. Otherwise, to me, I think the latter information from lines 336-341 is necessary to explain why the IS-CCP simulator produces more clouds in the Arctic summer that the CLARA simulator as seen in Fig. 7

Reviewer 2: L352. The trends calculated in this section are not decadal trends. I believe that what you are trying to say is that they are trends over the entire record, expressed in units of %/decade

Answer: Well, yes, this is what we are saying. We are using the wrong notation here and have fixed the unit and description

Reviewer 2: - L354. Please use the correct units (%/decade). Same for figures 8 and 9. I would even suggest to change the units to 1/decade, as changes in %

can lead to confusion in its interpretation (absolute percent change vs relative change).

Answer: We have fixed the units, but instead of changing the unit to 1/decade, we decided to keep (%/decade) and describe clearly in the text that the trends are trends in an absolute sense.

Answer: OK

[revised manuscript text omitted]

---

## Author Response (AR2)

Dear Editor,

Thanks again for taking the time to assess our article. Below are my point by point answers (in black) to the questions posed suggestions made (in blue). Together with these answers, I have also sent a marked up version of the updated article to Copernicus. We made some more changes to the grammar of the text in some places and we also moved around the arguments in the conclusion section for clarity's sake.

Thank you
Salomon Eliasson

P 2, L 41: ... based ON geostationary... OK, done
P 6, L 160: ... than OVER most continental... OK, done
P 8, L 212-213, sentence "First, the simulator ... ". Please reformulate. It is not clear what you mean (I was guessing but unsure whether my guess was correct). OK, done
Next line: correct brackets around Eliasson... OK, done
P 9, L 233, 234: you can simply write "cloudy subcolumns... are treated as being cloud-free and the subcolumns above this ... ". The "100%" should be avoided or replace by "all". OK, done
P 11, L 267: delete green text. OK, done
P 15, L 331: correct panel label. OK, done
P 18, L 364-365: The sentence is incomplete. OK, done

[revised manuscript text omitted]